# Telemedicine System Applicability Using Drones in Pandemic Emergency Medical Situations

Paul Lucian Nedelea [1], Tudor Ovidiu Popa [1,*], Emilian Manolescu [1], Catalin Bouros [1], Gabriela Grigorasi [1], Doru Andritoi [2,*], Catalin Pascale [3,*], Avramescu Andrei [3] and Diana Carmen Cimpoesu [1]

1 Emergency Medicine Department, Faculty of Medicine, University of Medicine and Pharmacy "Grigore T. Popa Iasi", 700115 Iasi, Romania; paul.nedelea@yahoo.com (P.L.N.); manolescu.emilian@gmail.com (E.M.); bouroscatalin@yahoo.com (C.B.); gabriela.tiulica@yahoo.com (G.G.); dcimpoiesu@yahoo.com (D.C.C.)

2 Biomedical Science Department, Faculty of Bioengineering, University of Medicine and Pharmacy "Grigore T. Popa Iasi", 700115 Iasi, Romania

3 INCAS—National Institute for Aerospace Research "Elie Carafoli", 061126 Bucharest, Romania; avramescu.andrei@incas.ro

* Correspondence: tudor.popa@umfiasi.ro (T.O.P.); andritoi.doru@umfiasi.ro (D.A.); pascale.catalin@incas.ro (C.P.); Tel.: +40-745-389-258 (T.O.P.)

**Abstract:** Drones have evolved significantly in recent years, acquiring greater autonomy and carrier capacity. Therefore, drones can play a substantial role in civil medicine, especially in emergency situations or for the detection and monitoring of disease spread, such as during the COVID-19 pandemic. The aim of this paper is to present the real possibilities of using drones in field rescue operations, as well as in nonsegregated airspace, in order to obtain solutions for monitoring activities and aerial work in support of the public health system in crisis situations. The particularity of our conceptual system is the use of a "swarm" of fast drones for aerial reconnaissance that operate in conjunction, thus optimizing both the search and identification time while also increasing the information area and the operability of the system. We also included a drone with an RF relay, which was connected to a hub drone. If needed, a carrier drone with medical supplies or portable devices can be integrated, which can also offer two-way audio and video communication capabilities. All of these are controlled from a mobile command center, in real time, connected also to the national dispatch center to shorten the travel time to the patient, provide support with basic but life-saving equipment, and offer the opportunity to access remote or difficult-to-reach places. In conclusion, the use of drones for medical purposes brings many advantages, such as quick help, shortened travel time to the patient, support with basic but life-saving equipment, and the opportunity to access remote or difficult-to-reach places.

**Keywords:** drones; emergency medicine; unsegregated space flight; telemedicine; first aid; COVID-19

## 1. Introduction

It is said that we live in exponential times, in which we advance in leaps and bounds. We all feel that the world is changing at an accelerated pace, and we can even observe the phenomenon on different levels of collective and individual life.

We are already surrounded by artificial intelligence, from autonomous vehicles and drones to virtual assistants and software that translates in real time, invests in the stock market, and contributes to the discovery of new drugs or algorithms capable of predicting our interests and social behaviors. Digital technologies regularly interact with the biological world, leading to a symbiosis between microorganisms, our bodies, the products we consume, and even the buildings in which we live.

Drones are some of the most interesting and sophisticated toys of the 21st century. Professionals have found ways to use drones to accomplish some previously impossible

tasks, while those looking to have fun have found in drones a toy that gives them a rewarding experience. In the context of increasingly sophisticated systems and growing requirements, the costs and risks associated with missions performed with manned aircraft on board, both material and human, can be very high. On the other hand, unmanned aircraft systems (UAS) provide real-time or near-real-time continuous and long-lasting information support without endangering the physical integrity of the aircrew.

Such types of systems can be used both in peacetime and in situations of armed aggression, in case of a state of siege, in calamity situations, or during the planning, management, coordination, control, and evacuation stages in case of accidents. Therefore, the involvement of these types of vehicles in operations related to the support of medical services and the delivery of consumables and medicines provides a good premise in order to develop and create logistics networks for drones.

Since their inception, drones have been used in various ways to monitor areas of natural disasters (e.g., earthquakes, floods, or fires) or areas of high biological danger. More recently, with the 2020 pandemic, other applications of medical drones have been identified with great potential to increase the efficiency of the system and save more lives, allowing the provision of humanitarian aid in affected areas and emergencies that require an effective response time [1,2]. Another use is to reduce the time it takes to deliver samples and laboratory products to remote health centers [3]. Vital parameters can be assessed in real time using two-way communication, facilitated by drone technology. Currently, even humanitarian aid organizations are using drones to cost-effectively deliver vaccines, pharmaceuticals, and medical supplies to remote locations or difficult-to-reach areas [4–8].

The main goal of the project was to implement a system in order to help medical teams in specific activities in conditions of calamity or accident. Drone-type flying devices were used as a support element both in terms of obtaining information related to the situation and also in the transport of equipment and materials of a medical nature such as defibrillators, oxygen, etc. Another idea led to the implementation of a system composed of a "swarm" of drones, each with a well-defined role, able to communicate both with the command center and between them, thus obtaining efficient interoperability.

## 2. Materials and Methods

We proposed the development of an innovative UAS system that will ensure the combined operation of several air and ground vectors as medical support elements, adapted particularly to combating the COVID-19 pandemic (through specific monitoring and surveillance actions, transport of medical materials, decontamination actions, etc.), with coordination from the command/support center in the test area.

In order to operationalize a structured system of identification, characterization, and intervention decisions using UAS for activities within the public health system within this project, a collaboration between "Elie Carafoli" National Aerospace Research and Development Institute (INCAS), Bucharest; Military Technical Academy Ferdinand I; University of Medicine and Pharmacy "Grigore T. Popa", Iaşi; and Iaşi Clinical Hospital of Pneumology and Mira Technologies Group was necessary.

System components are represented by multicopter platforms for control, search, surveillance, and transport with larger payload but slower speed, all of them being connected to a command and control center (Figure 1).

The main features of the UAV system components are listed below.

*Free multicopter platform*: This is an aerial platform with vertical take-off and landing, equipped with 6 electric motors for take-off/landing. The optical system on board the platform is a dedicated system that allows images to be taken for specific surveillance applications. The optical system consists of an EO/IR image sensor, a gyrostabilized turret, and a video transmitter. The maximum flight autonomy is 20 min.

*Multicopter transport platform*: This is an aerial platform with vertical take-off and landing, equipped with 4 electric motors for take-off/landing with the possibility of high

transport (max. 10 kg). The on-board optical system is a dedicated navigation phase monitoring (FPV) system. The maximum flight range is 20 min.

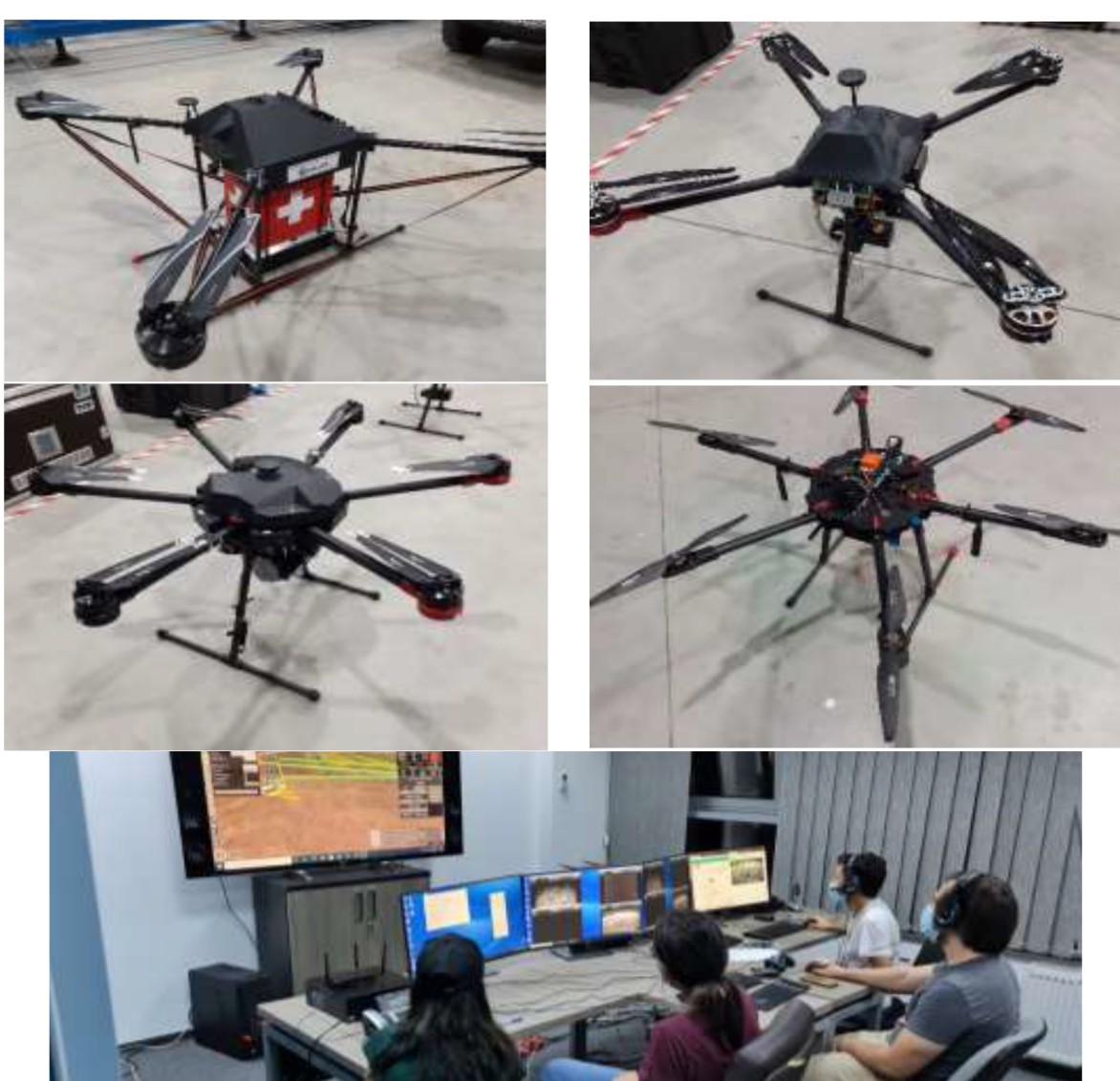

**Figure 1.** System components and concept of operation in 19 Sol/UAV.

*RF relay multicopter platform*: This is an aerial platform with vertical take-off and landing, equipped with 6 electric motors for take-off/landing. The main role of this air vector is to ensure the BLOS operation of the air vectors in the UAS system. The main process sensor is a MIMO modem with a working frequency of 5.8 GHz (Microhard). The maximum flight range is 30 min.

*Captive multicopter platform*: This is an aerial platform with vertical take-off and landing, equipped with 4 electric motors for take-off/landing with power from a ground cable. The on-board optical system is a dedicated image capture system for specific surveillance applications. The optical system consists of an EO/IR image sensor, a gyrostabilized turret, and a video transmitter. It allows the boarding of radio relay equipment (in the absence of an EO/IR sensor). The maximum flight autonomy is 24 h.

A basic element of the system is the mobile command/control point:

The ground control system is designed to ensure the flow of command and control between the operator and the drone. The control panel consists of a computer system for displaying the graphical interface with the operator and the software for flight monitoring.

It includes two elements: the interface, used to control UAVs, and the video analysis system, designed to capture and analyze information in the area of action.

The control panel meets the following requirements:

- Ensures the control and communication command functions in optimal conditions with minimum effort.
- Stores data for at least one mission.

*Dispatching System*

The system dispatcher ensures a superior level of coordination of the actions undertaken with the help of the UAS system (GCS + air carriers) (Figure 2). It consists of a series of hardware components for interfacing with the system (PC, standard monitors, 4K high-resolution monitor) and communication (4G GPRS modem, bidirectional audio communication equipment, telephone exchange). Attached to the dispatching point is a technical room that allows the accommodation of system operators before the operational phase. The staff serving the dispatching point consists of the operations commander, the medical manager, the secretarial operator, and the IT system engineer, all being involved in the efficient functioning of the system (Figure 3). The specific universal ground control software (UGCS) for unmanned vehicles was used to implement the specific control elements to ensure the operational functionalities of the air vectors in a collaborative regime. It allows communication and control with the best-known autopilots used in the field of UAV (PX4, Ardupilot, Micropilot, DJI, Yuneek, etc.). The software component (UGCS) that allows simulant management for multiple UAV systems is supported by the specific hardware component for communication. The communication system used is unitary. Multi-input multi-output (MIMO) modems are used in the 5.8 GHz band (Microhard with 128-bit AES encryption) configured in bridge mode (the AP arranged on GCS can communicate simultaneously with all modems embedded on the system's UAVs in the fleet).

We tested the system in a scenario that involved victims of an accident in a remote area. The UAV mobile dispatch deployed the RF relay drone and the leader drone for search activity, and after that, the transporter drone was sent to the place of the accident to provide medical assistance. The witnesses from the place of the accident interacted via audio and video with the medical dispatch, and they applied first-aid measures using the equipment from the drone.

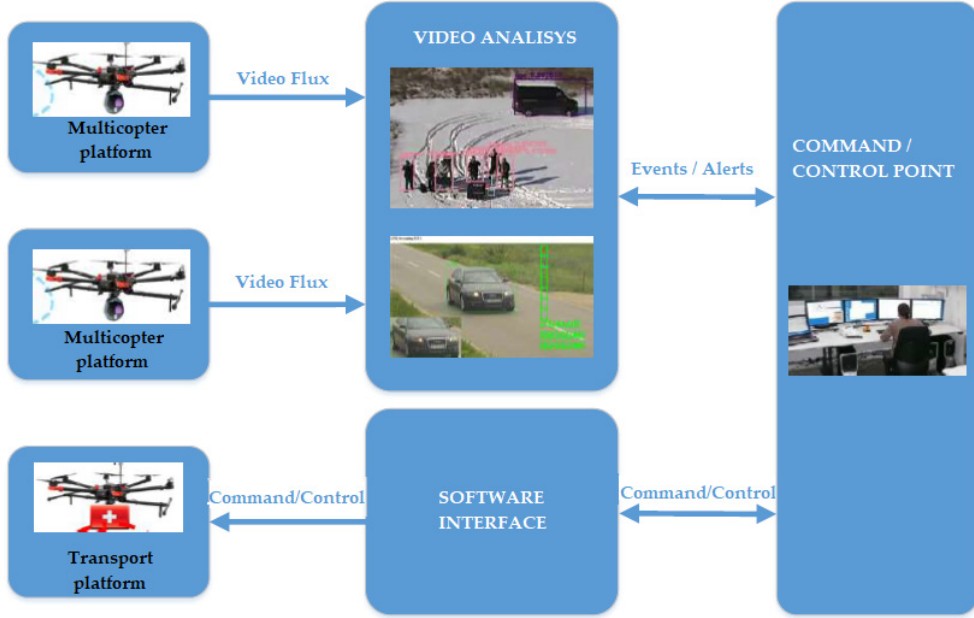

**Figure 2.** Software component of 19 Sol/UAV.

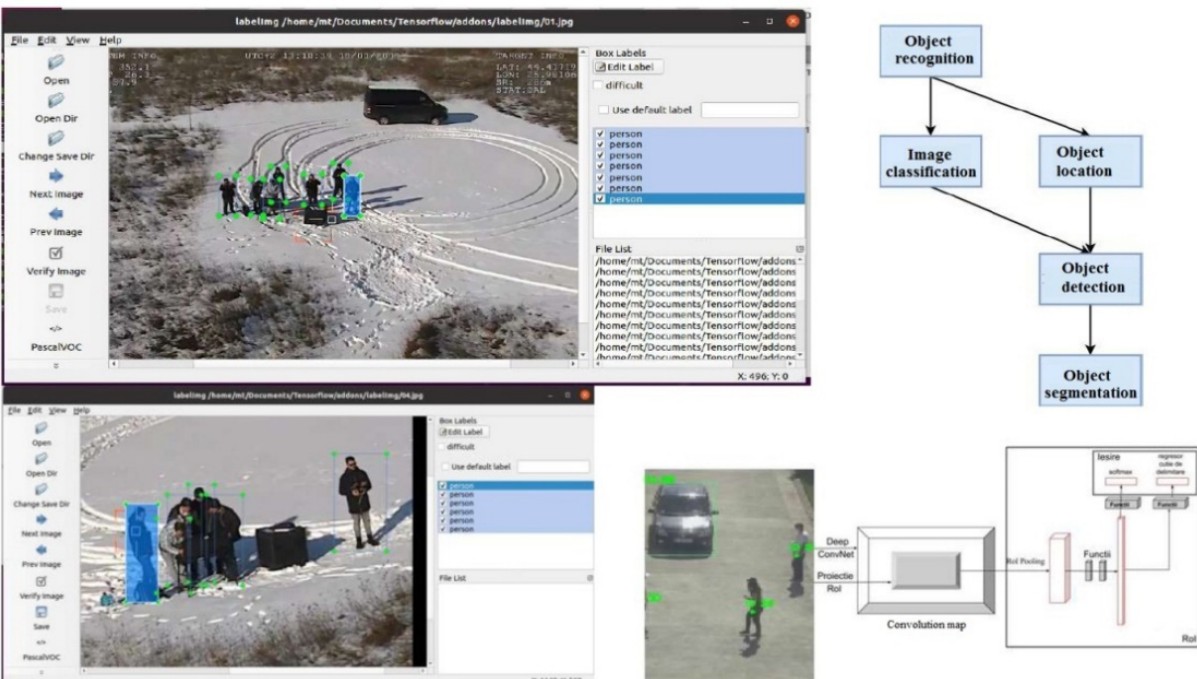

**Figure 3.** Chain of data acquisition in 19 Sol/UAV.

The implementation and operational testing of the system were carried out by the partners from INCAS at their experimental base in Strejnicu. This location was chosen taking into account the specifics of the area, near the city of Ploiesti and near the Strejnicu aerodrome (Gheorghe Banciulescu aeroclub), these factors being decisive for testing a UAS system of the type developed in this project (operation in airspace with heavy traffic, communication with the aerodrome control tower, operation in the vicinity of urban agglomerations) in real conditions.

## 3. Results

Our project 19 Sol/UAV was created based on our knowledge combined with international experience in telemedicine and drone utilization in remote areas.

We tested the system using a scenario that simulates an accident in a remote area, involving participants without any medical background.

The major phases of the test were:

(a)   The call of the person in difficulty was received by the dispatcher.

(b)   The dispatcher operator (operations commander) analyzed the request and arrangement of ground support elements (GCS, air carriers, and process payload/medical equipment) and decided to launch the intervention order with the advanced air vector (investigation vector) to the commander's mobile CC position.

(c)   The commander of the mobile CC station received the request, analyzed the flight conditions, and decided to launch the investigation vector (the launch of the intervention vector was made in less than 5 min from the receipt of the request of the person in difficulty). As a secondary technical phase, when the advanced vector reached halfway to the objective, the commander of the CC station launched the second drone, the RF relay drone, to ensure the BLOS connection and announced it to the dispatcher.

(d)   After the preliminary supervision of the objective with the advanced vector (and taking into account the information transmitted in parallel by the person in difficulty taken over by telephone by a medical operator), the operations commander decided to launch the transport vector with the mission to transport medical equipment to the person in difficulty. Once landed near the victim, the witnesses from the place of the accident were instructed on how to use the different medical devices and materials from the cargo of

the drone, guided step by step, in real time by the medical dispatch team, connected to an audio/video feed of the drone through the drone's mobile dispatch unit.

(e) They were able to assess the vital signs of the patient, use a defibrillator, a glucose test, a pulse oximeter, and an automatic sphygmomanometer, and they also correctly stopped severe lower limb bleeding using the medical equipment from the transporter drone.

(f) The commander of the mobile CC station received the request, withdrew the forwarded vector, and launched the transport vector with the payload on board.

(g) After the landing of the transport vector and the delivery of the medical equipment, the commander from the mobile CC station launched and recovered the transport vector and the relay vector at the base.

The main goals to be achieved by our system are the following:

- Development and operationalization of a specific class of UAS vehicles capable of aerial work in segregated and nonsegregated space, with advanced sensing capabilities, communication, and dynamic organization in operational networks;
- Implementation of a distributed management system for dynamic UAS vehicle networks performing aerial work in a nonsegregated space;
- Operationalization of a structured system of identification, characterization, and intervention decision using UAS systems for activities within the public health system;
- Adaptation of the operating system to the conditions imposed by the crisis situations, e.g., COVID-19 pandemic; ensuring support capacity at the national level for action and intervention in crisis situations.

In order to analyze the images acquired by drones in the field, a series of algorithms were developed to identify people and vehicles within a range, as well as features related to the health of the subjects present and compliance with quarantine rules (e.g., COVID-19). One of the algorithms developed in order to achieve these goals is illustrated in Figure 4.

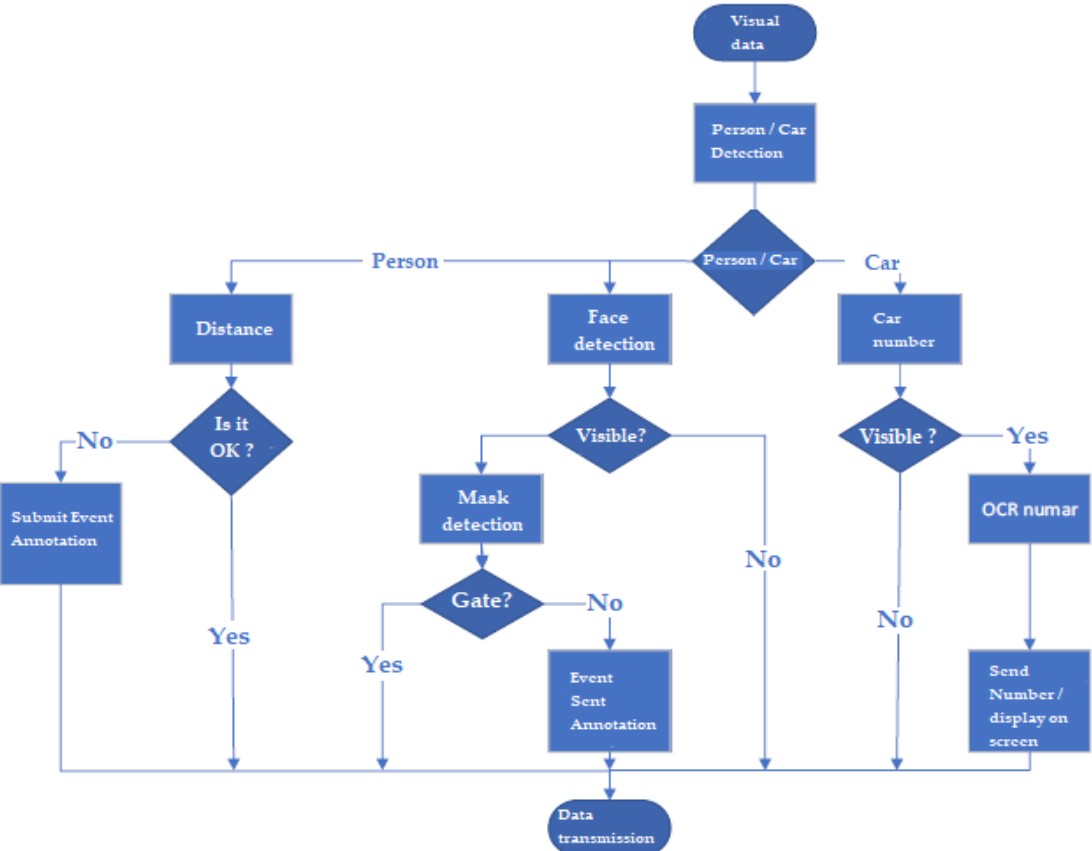

**Figure 4.** Operational algorithm of 19 Sol/UAV.

The UAS software component must provide the necessary functionality to ensure the software functionalities necessary for the dispatcher (displaying the system components on the map and implementing the operating procedures specific to the medical factor), a software interface for process payload (bidirectional communication audio/video equipment), and, in particular, support actions to identify the degree of compliance with the rules of social distance (formation of human groups, analysis of road traffic, etc.).

Video analysis was performed for the detection of objects (people or machines), after which the distance between them was determined, i.e., the statistical density per unit of measurement, using an application from the field of machine learning (ML) that gives systems the ability to learn and improve their decisions automatically from experience, without being programmed. ML focuses on developing programs that can access data and use them for self-learning.

*Custom Training for Object Detection*

In video analysis based on artificial intelligence, the success of detection is given by two elements: the model used for detection, but it must be taken into account that a model of high accuracy also consumes large computational resources, and the data set used for training (thousands of annotated images must be generated with as much variety as possible in order to train a model).

In the scenarios of our project, the video images are taken from the UAV, from a higher height (50–100 m), and the person must be recognized by the algorithm with a top view. This is an unforeseen situation in regular training processes. Data sets made available under the "open-source" license provide images with people in front, side, back, or portrait view. In order to improve the detection process of platforms to identify the degree of compliance with the rules of social distance, it is necessary to create a data set with images filmed by UAVs and then used in training the object detection model.

In order to speed up the training process, a much more efficient method called knowledge transfer is used. Knowledge transfer learning is a machine learning method in which a model developed for a task is reused as a starting point for a model in a second task.

Two object detection algorithms were used, namely SSD_MobileNet_V2 and Faster_RCNN. Faster R-CNN models are more suitable for cases where high accuracy is desired and latency has a lower priority. However, if processing time is the most important factor, SSD models are recommended.

The set of images relevant for training were annotated by identifying, marking, and labeling each object of interest (people, cars in this case), after which the files necessary to drive the algorithm (train.csv, train.record, test.csv, test.record) were made. These files are the input data for training and evaluating the algorithm. We used TensorFlow as an end-to-end platform for building and implementing machine learning models.

TensorFlow offers several levels of abstraction. Building and preparing models using the high-level Keras API facilitates the TensorFlow and machine learning process, allowing us to easily drive and deploy models, regardless of the programming language or platform used.

Tensorflow uses protobuf files to configure the training and assessment process. At a high level, the configuration file we used was divided into 5 parts:

1. Model configuration: This defines what type of model will be trained (e.g., meta-architecture, function extractor).
2. Train_config, which decides which parameters should be used to drive the model parameters (i.e., SGD parameters, input preprocessing, and extractor initialization values).
3. Eval_config, which determines what set of values will be reported for evaluation.
4. Train_input_config, which defines on which data set the model should be trained.

Eval_input_config, which defines on which data set the model will be evaluated. Usually, this should be different from the training data set.

In Figure 5 are presented some captured images from the video recognition phase.

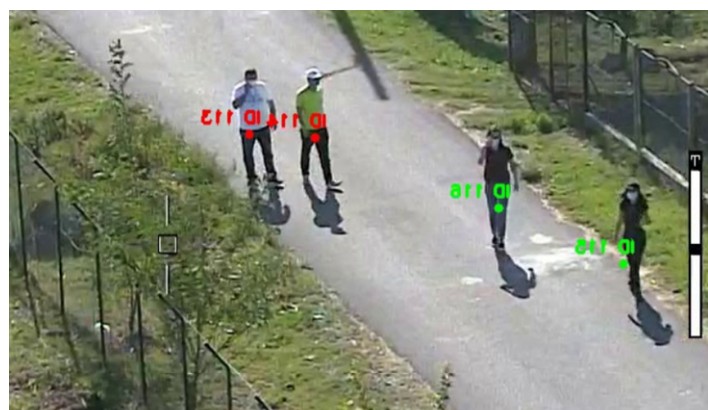
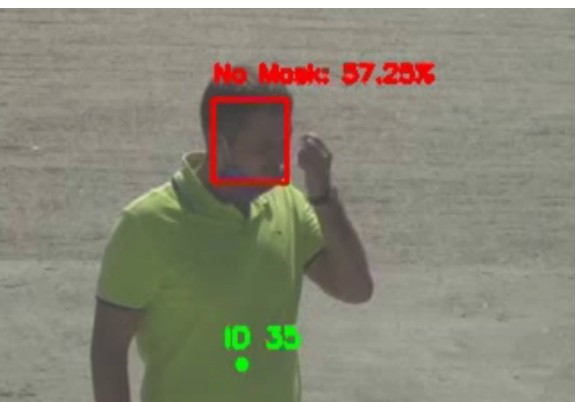

**Figure 5.** Video recognition (crowd area identification and facial mask presence).

Another particularity of the system is represented by the telemedicine audio/video capabilities in real time, materialized by ensuring a direct connection between a person who initiates an emergency call and a medical staff present in the dispatching office for the coordination of the system's activities at the national level. As a result of this facility, a quick intervention decision can be made by providing medical equipment that can be used to stabilize the condition of the person in need until the arrival of an intervention crew, for the provision of medical equipment or for transport of biological samples, in order to eliminate the need for direct access of personnel in the specific area.

The structure of the implemented system is illustrated in Figure 6.

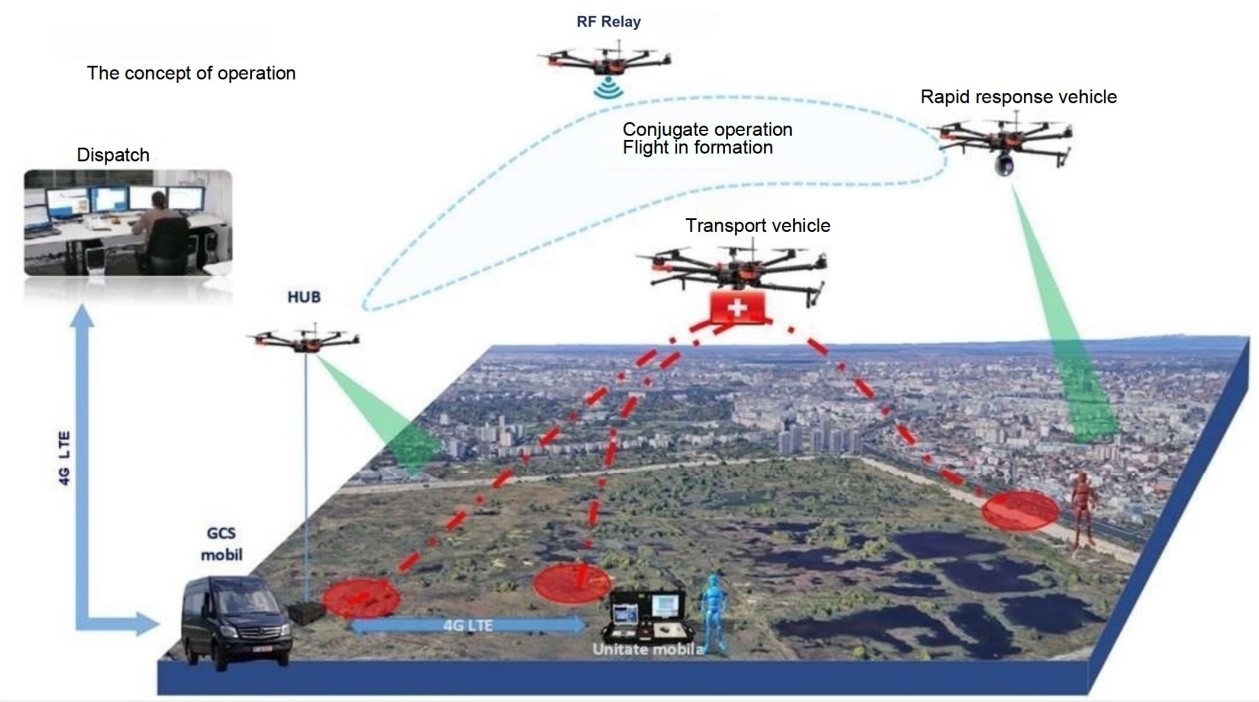

**Figure 6.** System components and concept of operation.

The UAS system within the developed application consists of four air carriers, each with a well-established function:

- Leading multicopter platform: Aerial platform with vertical take-off and landing, equipped with six electric motors for take-off/landing. The optical system on board the platform is an elaborate dedicated system that allows retrieval of images for



specific surveillance applications. The optical system consists of the following: image sensor EO/IR, gyrostabilized turret, bidirectional audio communication device, display device, and video transmitter. The maximum flight autonomy is 20 min.

- Quadcopter transport platform: Equipped with 4 electric motors for vertical take-off/landing, with the possibility of carrying a load of a maximum of 10 kg of medical equipment transport system (e.g., defibrillator) and automatic display device information (preprogrammed monitor). It was used to transport medical equipment. The on-board optical system is a dedicated navigation phase monitoring (FPV) system. The maximum flight range is 20 min.

- RF relay hexacopter platform: Equipped with six electric motors for vertical take-off/landing. The main role of this aerial vector is to ensure BLOS operation of air vectors in the UAS system. The main process sensor is a 5.8 GHz working-frequency MIMO modem (Microhard). Flight autonomy is maximum 30 min. Payload is represented by a Microhard MIMO modem (frequency 5.7250–5.8750 GHz; modulation ODFM/QPSK/16QAM/64QAM; standard radio 802.11a/n; emission power 6 dBm–30 dBm; channel bandwidth 20/40 MHz; detection ARQ error; WEP, WPA (PSK) encryption, WPA2 (PSK), WPA + WPA2 (PSK), AES). This helps to expand the signal coverage to ensure better transmission.

- "Captive" relay quadcopter platform: Consists of a drone powered by cable energy from the ground, with 2 h flight time resources, with functions that can be adapted according to the mission profile (RF relay, video surveillance). It is an aerial platform with vertical take-off and landing, equipped with four engines electrical for take-off/landing with power from a ground cable. The optical system on board the platform is a dedicated image capture system for specific surveillance applications. The optical system is composed of the following: EO/IR image sensor, gyrostabilized turret, and video transmitter. It allows boarding of equipment radio relay type (in the absence of the EO/IR sensor). The maximum flight autonomy is 24 h.

Table 1 shows the main technical characteristics for each component of the UAS system.

**Table 1.** Technical characteristics.

| Drone Function | Autonomy | Speed | Video Signal Delay | Altitude Maximum Accuracy | Level of Accuracy RTK Off/On | Loading |
|---|---|---|---|---|---|---|
| Leader | 20 min | 15 m/s | 4 ms | 50 m | 0.5 m/1 cm | 2 kg |
| Transporter | 20 min | 12 m/s | 4 ms | 50 m | 0.5 m/1 cm | 10 kg |
| RF relay platform | 30 min | 15 m/s | 4 ms | 50 m | 0.5 m/1 cm | 2 kg |
| "Captive" relay platform | 24 h | na | 4 ms | 50 m | 0.5 m/1 cm | na |

If necessary, it is possible to choose to add new units to the system in order to increase the area of air coverage and communication or to transport an increased amount of medical materials or equipment.

The UAV Control and Command Center (CCCUAV) integrates telemedicine techniques and uses procedures and recommendations to maintain the essential functions of the mission (s) during the event (Figures 7 and 8).

The plan sets out procedures and actions during the test that should have been facilitated to facilitate the operation and reduce the effects of the COVID-19 virus on the operational capacity of the CCUAV.

The procedures and actions presented in this plan were developed based on the recommendations of the Centers for Disease Control and Prevention (ECDC, CDC). The CCCUAV decision maker should coordinate operational decisions and actions with the rest of the agencies and should participate in the structure of the local control system.

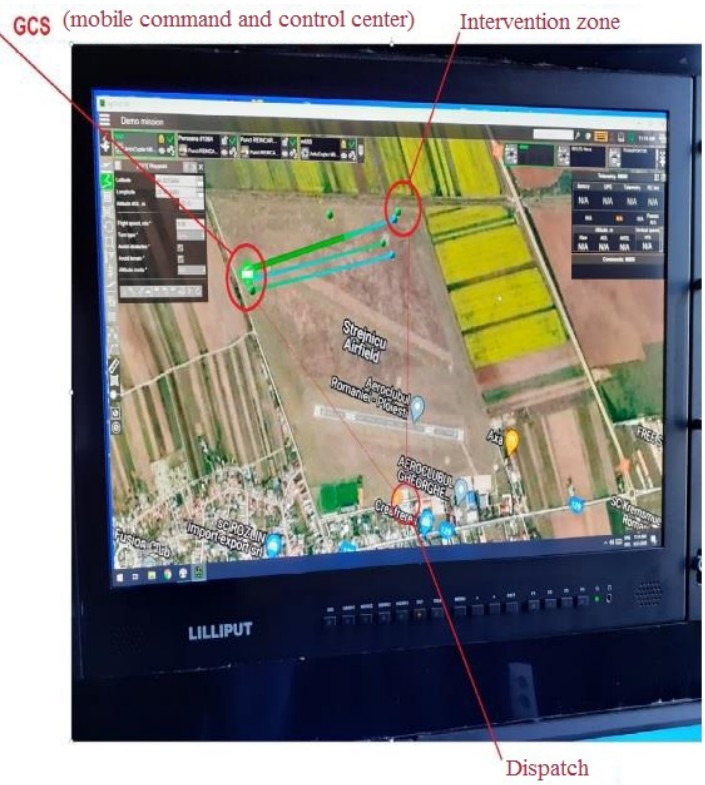

**Figure 7.** 19 Sol/UAV project test.

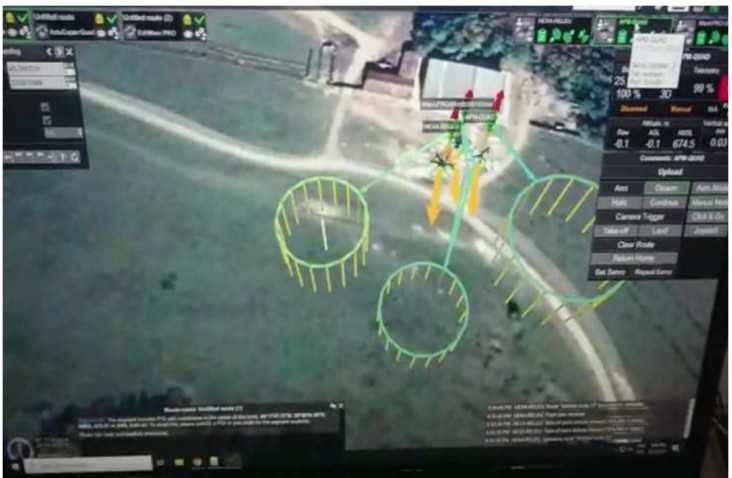
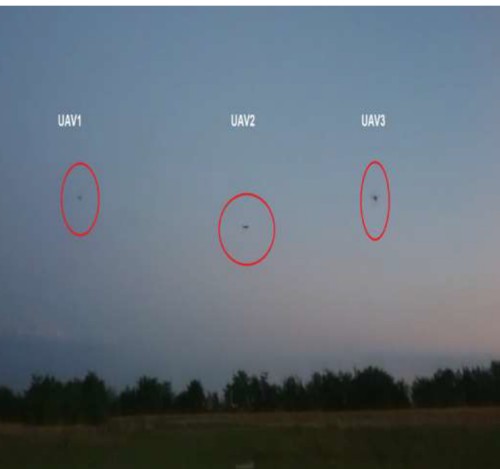

**Figure 8.** Implementation of specific control elements to ensure the functional operation of air vectors in a collaborative regime: testing 19 Sol/UAV solution.

## 4. Discussion

Researching the literature, we did not find published data about systems that include all of the characteristics of the system we presented, such as flight in segregated and nonsegregated spaces, advanced sensing capabilities, communication and dynamic organization in operational networks, and the possibility of delivering essential medical equipment.

The drones used in our project include capabilities of carrying medical devices and supplies necessary for emergency aid and integrating telemedicine capabilities. By integrating these capabilities into one aerial vehicle, we will be able to simultaneously provide diagnostics tools to medical personnel and some basic first-aid medical materials to victims. One novelty is the capability of using audio and video interactions with the persons in

need in real time (4 ms delay), giving the medical team a better image of the medical status of the patients.

The health services offered through telemedicine are:

- Synchronous, when a live, interactive conference is held;
- Asynchronous, which stores information and then can reproduce it;
- Monitoring patients with acute or chronic diseases [9–16].

For example, the UAV system can provide synchronous services and monitoring of patients diagnosed or suspected of having COVID-19 infection.

Personnel involved in UAV maintenance procedures will be required to be informed of the current epidemiological situation of COVID-19, including known risk factors for infection, symptoms, and clinical signs of COVID-19, recommended personal protective equipment (protective equipment), and decontamination procedures in accordance with national and international recommendations.

The CCCUAV will respond to requests directly or through government agencies to monitor the clinical status of people infected or suspected of being infected with COVID-19 and to collect biological samples or deliver medical equipment or therapeutic products they need when necessary.

The clinical status of patients will be assessed by remote monitoring procedures using the telemedicine technique. The evaluated parameters will include peripheral saturation in $O_2$, frequency of peripheral pulse, temperature, respiratory rate, blood sugar level, and general clinical examination/patient inspection using the integrated UAV camera and medical device kit transported by drones [13–18].

Some of the monitored parameters will also involve the input of the patient/relatives (use of the glucometer, application of the pulse oximeter, application of the monitoring/defibrillation electrodes, and use of the automatic external defibrillator).

The UAV monitoring sensors will ensure the real-time transmission of the main biometric data of the patient to the CCCUAV and would make it possible to transmit to the patient the emergency medical indications from the specialist doctors (in the dispatcher or from the emergency reception units) [19–21].

Achieving these goals involves dispatching and using telemedicine.

The large number of people living in remote areas, away from specialized medical centers, increasing the life expectancy of the population, the small number of emergency crews available in the hospital consisting of a doctor, lack of adequate road infrastructure, and the need for continuing medical education medical staff are key elements that have supported and formed the basis for the development of telemedicine systems. More recently, the need to use these techniques to prevent direct contact between patients with highly contagious pathology and medical staff has been added [17–21].

Over time, due to technological progress, the interconnection of the population through electronic devices, the desire for the comfort and convenience of people, and the trend of integrating telemedicine into as many medical specializations as possible has arisen. The expansion of telemedicine networks is based on the idea of providing access to health and medical education to a large population of patients [16–21].

Another approach we have in mind regarding this system is the application of telemedicine systems in case of natural disasters.

Natural disasters, such as Hurricanes Harvey and Irma, caused unprecedented severe flooding in 2017 in southeast Texas and off the coast of Florida (USA). During these catastrophes, the telemedicine systems of several hospitals provided free teleconsultations for victims [21–23].

The Doctor-on-Demand platform offered in August–September 2017 teleconsultations through the application or by accessing the company's website. Doctors could diagnose, prescribe, and refer patients to other departments, such as the emergency department, if there was a life-threatening condition [24–26].

The peculiarity of this study was determined by the similarity of the diagnoses made to the victims of the hurricanes with the diagnoses of the same period made to the patients

in areas unaffected by disasters who requested a teleconsultation. Although there has been a high demand for the use of telemedicine, the study suggests that telemedicine does not currently have a strong impact on medical problems caused by natural disasters but rather on the rudimentary needs of patients [15–20,24–26].

Emergency medicine has the potential to provide patients with rapid help through telemedicine, especially in cases where delayed treatment negatively affects healing without sequelae. A pathology that has proven the importance of every minute in diagnosing and initiating treatment is acute myocardial infarction with ST segment elevation (ST elevation in English), where the recognition of changes highlighted in electrocardiography can lead to a faster and improved intervention, clinical recovery of the patient, limited area of myocardial necrosis, preservation of the ejection fraction, and decreased mortality [21].

Telemedicine has also proven its applicability in cases of natural disasters, where the shortage of specialized medical staff has been ameliorated through teleconsultations, an example being the use of e-Health in 1988 in the case of the devastating earthquake in Armenia, which had a magnitude of 6.9 degrees on the Richter scale [15–19].

This earthquake was soon followed by another, with a magnitude of 5.8 on the Richter scale, and because both took place during lunchtime, there were about 55,000 dead and over 130,000 injured [15–19].

A common trend of the 21st century is to centralize specialized surgical and medical care in densely populated areas, but there are still more and more territories with a shortage of qualified medical staff, becoming a difficult barrier for those living outside densely populated areas to overcome the complex services they need. The most feared aspect of the centralization of overspecializations is the possibility of depriving a considerable number of people of quality medical services. In addition, centralization has made high-quality services less accessible to discriminated populations, those outside the metropolitan areas, and areas with low socioeconomic status [27–33].

*Ethical Considerations on Telemedicine*

The fields of application of telemedicine are represented by the provision of medical care and monitoring of patients in the territory, in medical clinics that do not have medical staff specialized in diagnosing and treating a certain pathology, as well as scientific research, medical education or medical management, natural disasters, preventive medicine, and mobile applications [34–38].

The main ethical issues in the field of telemedicine are related to the nonperformance of the classic clinical examination, the loss of direct communication between doctor and patient, thus causing the concept of trust between the two people involved, the transmission of incomplete information, to become implicitly insufficient. Specialization, maintaining the confidentiality of the personal data of those who have benefited from telemedicine services, and ensuring the cyber security of the patient's medical information (misuse by unauthorized persons of personal information may compromise the principles of storing people's private medical information).

Our system limitations were represented by the payload vs. autonomy factor and the challenge encountered by drones during flight and in the moment of search for victims in some particular weather or accident-related flight conditions, described also by another researcher [39]. In addition, we encountered some challenges because of the bird's presence in the flight test area. During one of the tests, one drone hit a pigeon in flight, but it was able to finalize the mission, helped by our redundant technology approach. We propose for the future to also improve the autonomous flight capability [40].

Another limitation we are aware of is the identification of people/victims in an environment that limits the use of video sensors used by drones. A successful solution could be the implementation and adaptation of a light detection and ranging (LIDAR)-type system for their identification. At the moment, there are projects that use drones equipped with LIDAR technology but for remote inspection of industrial structures, which presents risks for human health [41].

## 5. Conclusions

Drone technology can improve patient survival, outcomes, and quality of life, especially for patients living in remote areas that lack infrastructure.

Drones are a promising option for improving the management of patient work and quality of life, especially for remote or underfunded areas with poor infrastructure. Their cost savings compared to ground transportation, speed, and comfort make them an effective solution in the field of emergency medicine. To date, our preliminary results suggest that the use of drones in emergency situations may be feasible, as they are already increasingly accepted by the public. To make this potential drone-saving application a reality, more research is needed to put this practice into routine use as we navigate a changing regulatory environment, public safety and confidentiality, and public acceptance. As drones evolve at low cost, batteries that significantly extend the flight time and overall life of these aircraft, lighter frames with increased carrying capacity, and reorganization are required in the case of regulatory agencies to keep up with the rapid advances of this technology.

Drones can play a key role in helping in various ways to prevent the further spread of the COVID-19 outbreak by helping authorities in this regard. A solution we proposed refers to the use of this technology for aerial surveillance of individuals who do not comply with restrictions or do not know what the latest restrictions are.

In addition to surveillance, drones equipped with a two-way communication system can transmit messages and information useful to the medical process. By integrating these capabilities in one aerial vehicle, we are able to simultaneously provide diagnostics tools to medical personnel and some basic first-aid medical materials to victims.

In general, by using an increased number of drones, the response time in medical situations can be considerably improved. From our preliminary data, we also see a more efficient analysis of field data.

Since people were worried about contracting the infection, in order to avoid face-to-face contact, we thought about equipping drones with infrared cameras in the future, to measure the temperature of people remotely.

With the technology invested in the medical field, specialist doctors in large cities can provide professional care to patients in rural areas or other areas without full medical staff, without the need for transportation to the medical center and thus costs at the hospital level. Information can be obtained directly from the patient's living environment, long waiting periods can be eliminated, several people can be examined per day, and constant monitoring of critically ill patients can be ensured.

Technical advances in the near future can increase payload capacities, can increase flying distances, and will lead to the integration of drone networks into existing 112/911 and EMS systems.

In the context of the pandemic, direct contact and possible contamination of medical staff are also avoided, a real problem. The official statistics describing infection rate among medical staff varies between 9% and 10% of the total patients diagnosed with COVID-19.

**Author Contributions:** Conceptualization, P.L.N., T.O.P., D.C.C., C.P. and D.A.; methodology, P.L.N. and T.O.P.; investigation, P.L.N., T.O.P., D.C.C., C.P., E.M., C.B. and G.G.; resources, E.M., C.B., G.G. and A.A.; writing—original draft preparation, P.L.N., T.O.P., D.C.C., C.P. and D.A.; writing—review and editing, P.L.N., T.O.P., D.C.C., C.P., E.M., C.B., G.G., A.A. and D.A.; visualization, T.O.P. and D.A.; supervision, P.L.N., T.O.P., D.C.C., C.P. and D.A.; project administration, D.C.C. All authors have read and agreed to the published version of the manuscript.

**Funding:** This project was funded by UEFISCDI, National Plan for Research, Development, and Innovation for the period 2015–2020 (PNCDI III); Program 2: Increasing the Competitiveness of the Romanian Economy through Research, Development, and Innovation; Subprogram 2.1: Competitiveness through Research, Development, and Innovation—Solutions.

**Acknowledgments:** We wish to express our warm thanks to Radu Ciorap and to Nae Catalin for their support and to all the team members involved in the development and implementation of the project 19 Sol/UAV "Solutions and Systems for Monitoring and Aerial Work in Public Health System Support for the COVID-19 Pandemic using UAS Systems".

**Conflicts of Interest:** The authors declare no conflict of interest.

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
