# Peer review of "Telemedicine System Applicability Using Drones in Pandemic Emergency Medical Situations"

_electronics, doi:10.3390/electronics11142160_

Round 1
Reviewer 1 Report
The paper is supposed to present a multiple UAV based telemedicine system. I understand the system is conceptual and under development. But the paper does not even present the concept. It just very briefly introduces the hardware to be used and discusses lots of off-topic things. For example, in lines 196 to 199, is there any relationship between COVID symptoms and your system? Also, this paragraph needs proper citation references.
The paper doesn’t have any literature review of the existing telemedicine system. What’s the difference between the authors’ one and the existing one? Any benefits? Pros and Cons?. Novelty?
There is no result in the Result section as well. What is the level of fly accuracy? Speed? Flight duration? You are talking about video analysis and command points. I would expect you give some quantified results about video analysis accuracy and command delays, etc.
Section 4 should be part of the background introduction in Section 1. Discussion should be the discussions based on the experiment results presented in Section 3.
Lines such as 333, 348. 357 should be subtitles. Please read the system generated document again before pressing the submit button.
Author Response
Dear Reviewer
Thank you very much for your observations. We realized that we present our project results mostly from the point of view of medical personnel. But after reading your comments we did a step backward and revised again our work and hopefully we were able to improve the content of article.
The paper is supposed to present a multiple UAV based telemedicine system. I understand the system is conceptual and under development. But the paper does not even present the concept. It just very briefly introduces the hardware to be used and discusses lots of off-topic things. For example, in lines 196 to 199, is there any relationship between COVID symptoms and your system? Also, this paragraph needs proper citation references.
We improved the presentation of the concept and described better the functionality, the medical assistance and telemedicine role in our drones system.
The paper doesn’t have any literature review of the existing telemedicine system. What’s the difference between the authors’ one and the existing one? Any benefits? Pros and Cons?. Novelty?
We address this, and we describe the benefits of having diagnostics and first aid capabilities in one place, all together in one medical drone.
There is no result in the Result section as well. What is the level of fly accuracy? Speed? Flight duration? You are talking about video analysis and command points. I would expect you give some quantified results about video analysis accuracy and command delays, etc.
Indeed, working so many on this article we miss the most important part, the test in real condition we perform to assess the capabilities of our system. We include this in paper.
Section 4 should be part of the background introduction in Section 1. Discussion should be the discussions based on the experiment results presented in Section 3.
Yes , it is more logical like this, thank you for this observation, we did the changes.
Lines such as 333, 348. 357 should be subtitles. Please read the system generated document again before pressing the submit button.
We address this in text.
Thank you again for your time in helping us to improve our article.
Reviewer 2 Report
The topic under study in this manuscript is of cardinal importance. However, this manuscript suffers from some major shortcomings. Therefore, the authors should improve the quality of their manuscript by applying the following revisions:
1- The theoretical contributions of this study should be highlighted in the Abstract section.
2- The research objectives and contributions are still unclear in the Introduction section.
3- The features of this research, compared with the previous studies, should be presented by reviewing published papers in the field of the study. Also, the authors can demonstrate the research gap.
4- The authors should add the research methodology framework in Section 2.
5- The research limitations and development suggestions for future researchers should be provided in the Conclusion section.
Author Response
Dear Reviewer
Thank you very much for your observations. We realized that we present our project results mostly from the point of view of medical personnel. But after reading your comments we did a step backward and revised again our work and hopefully we were able to improve the content of article.
- The theoretical contributions of this study should be highlighted in the Abstract section.
We implement this.
- The research objectives and contributions are still unclear in the Introduction section.
We try to clarify this, our main goal was to create a multiplatform aerial vehicle, capable of helping persons in different medical emergencies, having not only diagnostics but also treatment capabilities.
- The features of this research, compared with the previous studies, should be presented by reviewing published papers in the field of the study. Also, the authors can demonstrate the research gap.
We improve this section in our text.
- The authors should add the research methodology framework in Section 2.
We describe the test, indeed this was something we miss to clarify better in our paper.
- The research limitations and development suggestions for future researchers should be provided in the Conclusion section.
We improved the Conclusion section.
Thank you again for your time in helping us to improve our article, and we hope it is much better now.
Round 2
Reviewer 1 Report
The manuscript is still like a product datasheet, not a professional research paper. It has to be rejected and massively revised.
The following questions are NOT addressed:
The paper doesn’t have any literature review of the existing telemedicine system. What’s the difference between the authors’ one and the existing one? Any benefits? Pros and Cons?. Novelty?
There is no result in the Result section as well. What is the level of fly accuracy? Speed? You are talking about video analysis and command points. I would expect you give some quantified results about video analysis accuracy and command delays, etc. You need to present real test results as well as theory values. Discuss what are the differences between, and why these are different.
Section 4 should be part of the background introduction in Section 1. Discussion should be the discussions based on the experiment results presented in Section 3.
Author Response
Dear reviewer,
Thank you kindly for your suggestions, hope that now we did a better job on improving our manuscript following your advices.
- The paper doesn’t have any literature review of the existing telemedicine system. What’s the difference between the authors’ one and the existing one? Any benefits? Pros and Cons?. Novelty?We hope we describe better the actual utilisation of telemedicine system in emergency medicine field in the discussions chapter. Also we stress in the manuscript the novelty of our concept, which include localisation of persons, comunications(audio-video) and the posibillity to deliver medical materials and instructions in real time about how to use specific medical equipment.
2. There is no result in the Result section as well. What is the level of fly accuracy? Speed? You are talking about video analysis and command points. I would expect you give some quantified results about video analysis accuracy and command delays, etc. You need to present real test results as well as theory values. Discuss what are the differences between, and why these are different.
We followed the advice and improved the results chapter with the data we had access to, because one of the members of our project is the Romanian Academy of Military Sciences and they developed "in the house" part of the technical equipment and that way some technical data are classified. We include now data regarding video analysis accuracy and command delays.
Also, in table 1 we input the data regarding technical characteristics of our drones.
3. Section 4 should be part of the background introduction in Section 1. Discussion should be the discussions based on the experiment results presented in Section 3.
We did the suggested changes in the manuscript, improving the Discussion and focusing more on our test and results.
We appreciate your help and we want to express our thanks again for taking the time to improve our manuscript!
Best regards,
Ovidiu Popa
Reviewer 2 Report
I appreciate the authors' efforts to consider my comments. Nevertheless, I think the analysis of the results section should be improved considerably.
Author Response
Dear Reviewer,
Thank you very much for the suggestions, we hope we have managed to improve the results chapter of our article with the data's we have access, because one of our partner is Romanian Military Technical Academy and some of the technical details were clasified.
We also made improvements in the rest of the article.
Best regards,
Ovidiu Popa
Round 3
Reviewer 1 Report
The paper is much better now. There are some format issues need to be solved. e.g. Line 455, Line 471.